# Effects of Combined Use of Olive Mill Waste Compost and Sprinkler Irrigation on GHG Emissions and Net Ecosystem Carbon Budget under Different Tillage Systems

**DOI:** 10.3390/plants11243454

**Published:** 2022-12-09

**Authors:** Damián Fernández-Rodríguez, David Paulo Fangueiro, David Peña Abades, Ángel Albarrán, Jose Manuel Rato-Nunes, Carmén Martín-Franco, Jaime Terrón-Sánchez, Luis Andrés Vicente, Antonio López-Piñeiro

**Affiliations:** 1Área de Producción Vegetal, Escuela de Ingenierías Agrarias—IACYS, Universidad de Extremadura, 06071 Badajoz, Spain; 2LEAF—Linking Landscape, Environment, Agriculture and Food Research Center, Associated Laboratory TERRA, Instituto Superior de Agronomia, Universidade de Lisboa, Tapada da Ajuda, 1349-017 Lisboa, Portugal; 3Área de Edafología y Química Agrícola, Escuela de Ingenierías Agrarias—IACYS, Universidad de Extremadura, Ctra de Cáceres, 06071 Badajoz, Spain; 4Instituto Politécnico de Portalegre, Escola Superior Agrária de Elvas, 7300-110 Portalegre, Portugal; 5Área de Edafología y Química Agrícola, Facultad de Ciencias—IACYS, Universidad de Extremadura, Avda de Elvas s/n, 06071 Badajoz, Spain

**Keywords:** methane, nitrous oxide, carbon balance, sprinkler, organic amendment, direct seeding

## Abstract

Traditional rice (*Oryza sativa* L.) production by flooding is a source of greenhouse gases (GHG), especially methane. The high consumption of water, as well as the chemical and physical degradation caused by these traditional practices in rice soils, is promoting a decrease in rice production in the Mediterranean area. The aim of this study was to monitor GHG emissions and the net ecosystem carbon balance (NECB) from rice produced with sprinkler irrigation techniques and also assess the impact of olive mill waste compost (C-OW) application and tillage on GHG emissions and the NECB. A field experiment for irrigated rice production was implemented by considering four different treatments: (1) tillage (T); (2) no tillage—direct seeding techniques (DS); (3) application of C-OW followed by tillage (TC); and (4) application of C-OW followed by direct seeding (DSC). The C-OW was only applied in the first year at a dose of 80 Mg ha^−1^. GHG emissions were monitored over three years in these four treatments in order to estimate the direct (first year) and residual (third year) effects of such practices. The application of C-OW caused an increase of 1.85 times the emission of CO_2_-C in the TC-DSC compared to the T-DS in the first year. It is noteworthy that the TC treatment was the only one that maintained an emission of CO_2_-C that was 42% higher than T in the third year. Regardless of the treatments and year of the study, negative values for the cumulative CH_4_ were found, suggesting that under sprinkler irrigation, CH_4_ oxidation was the dominant process. A decrease in N_2_O emissions was observed under direct seeding relative to the tillage treatments, although without significant differences. Tillage resulted in an increase in the global warming potential (*GWP*) of up to 31% with respect to direct seeding management in the third year, as a consequence of the greater carbon oxidation caused by intensive tillage. DS presented a positive NECB in the accumulation of C in the soil; therefore, it provided a greater ecological benefit to the environment. Thus, under Mediterranean conditions, rice production through a sprinkler irrigation system in combination with direct seeding techniques may be a sustainable alternative for rice crops, reducing their *GWP* and resulting in a lower carbon footprint. However, the use of C-OW as an organic amendment could increase the GHG emissions from rice fields irrigated by sprinklers, especially under tillage conditions.

## 1. Introduction

Today, rice (*Oryza sativa* L.) is a crop of enormous importance for feeding the world population in permanent growth. Being the second most cultivated cereal in the world, it is the main source of food for more than 50% of the world’s population [1]. In Europe, rice production takes place mainly in Mediterranean basin countries, with 70% of the total rice production attributed to Spain and Italy. In these countries, climate change has led to a sharp decrease in rainfall [2,3], jeopardizing the survival of the traditional crop, which is extremely important in environmental, economic and social terms. In fact, rice crops are one of the crops with the highest water consumption, being able, particularly in hot years, to reach values in the order of 35 000 m^3^ ha^−1^ [4]. It is therefore urgent to increase the efficiencies of water use [5,6], either through alternating flooding systems or through irrigation by sprinkling [7], and by either changing soil tillage systems or the use and typology of production factors.

In Mediterranean ecosystems, rice production coexists with other agroforestry crops typical of this region, namely olive groves. Thus, the olive oil extraction industry has a very significant socio-economic impact. However, important amounts of wastes are generated by this industry. In the particular case of two-phase continuous extraction, which is the commonest process used, the amount of waste represents about 80% of the total weight of fruit processed. Thus, from 1000 kg of olives, 800 kg of waste and 200 kg of oil are produced. Therefore, it is urgent to provide an economically, technically, and environmentally sustainable destination for this waste. This is a serious and emerging problem, and in the 2021/2022 campaign, several olive oil production companies, both in Spain and Portugal, had to stop working because they had no outlet for this residue. Olive oil production residues are, however, highly phytotoxic, containing high amounts of phenolic substances, lipids, and organic acids [8]. These characteristics make its application without any previous treatment on agricultural soils undesirable [9], and it is even prohibited by the legislation of several EU countries, namely Spain, Portugal, Italy, and Greece. On the other hand, this residue is very rich in organic matter and has some nutrients of interest for agricultural activity [8,10]. This product can be composted, with obvious benefits for agricultural use in terms of composition, namely a much lower content of phenolic compounds and a much more balanced C:N and C:P ratio [9,11,12,13]. The use of olive pomace compost as organic fertilizers in agricultural activity has numerous advantages [9,13], including increasing soil organic matter, which is a very scarce component in Mediterranean soils and of enormous importance for the sustainability of agricultural systems [9,13,14]. The use of this co-product of olive oil production as organic fertilizer in rice fields is an interesting possibility from an economic, agronomic, and environmental point of view that emphasizes the principles of a circular economy, using the by-products of agricultural activity as fertilizers in the same activity. However, it is necessary to carry out more studies that can clarify the advantages/disadvantages of the use of this residue as an agricultural fertilizer.

In order to respond to the new challenges of rice production, namely greater environmental sustainability, less available labor, and significantly reduced production costs [15,16], new techniques of production have been implemented, moving from a traditional tillage system to minimal tillage or direct seeding and using sprinkler irrigation or intermittent flooding [17]. Direct sowing is a cheaper system, with a faster implantation of the crop, less use of labor, a lower water consumption [18], a greater possibility of obtaining optimal seeding densities [17], and lower greenhouse gas emissions [19]. However, some negative aspects are sometimes pointed out for this technology, namely a greater number of weeds, which are more difficult to control; less support for rice plants; and, consequently, a lower production [20]. According to Tiefenbacher et al. [21], this shift should substantially reduce crop water requirements, weed biota, greenhouse gas emissions, and soil organic-matter turnover and improve nutrient relations and carbon sequestering in the soil.

Another concern regarding flooded rice crops is the huge production of greenhouse gases, especially methane and nitrous oxides, emitting four times more GHG than wheat or maize [22]. According to these authors, a normal rice crop is responsible for emissions in the order of 100 kg of CH_4_-C ha^−1^ season^−1^. Several strategies have been implemented to respond to this serious environmental problem, namely by changing the irrigation system, the mobilization system, the residue management, or even the fertilization management [22]. These changes in the production system have to be considered very carefully, since a reduction in methane emissions, achieved by irrigating rice crops without permanent flooding, is usually accompanied by an increase in nitrous oxide emissions [23,24]. Indeed, the application of organic fertilizers can have a significant effect on greenhouse gas emissions [25]. There are several techniques that can lead to an increase in carbon sequestration in the soil and thus reduce its presence in the atmosphere [26]. The first technique, which aims to decrease the mineralization of organic matter, is the use of reduce tillage or, even more efficiently, no tillage [27]. Another technique is the incorporation of organic matter into the soil, increasing carbon sequestration in the form of special stable organic residues [28]. However, the application of organic matter leads to an increase in the CO_2_ emissions produced during the mineralization process, and close attention must be paid to the balance between the amount of CO_2_ released and the amount of carbon sequestered in the soil, so that this balance does not lead to an increase in greenhouse gases in the atmosphere. This balance will obviously depend on the amount of compost added to the soil, the type of compost, and soil management practices [29]. Regarding the emission of nitrous oxides, the effect of applying organic compounds to the soil is not so clear, depending on the type of compound and especially its C:N ratio [30]. Jeong et al. [31] found that compost application can be a reasonable soil management strategy to reduce the impact of GHG emissions and to increase crop productivity in rice cropping systems.

The objective of this study was to monitor GHG emissions and the net ecosystem carbon balance (NECB) from rice produced with sprinkler irrigation and also assess the impact of the application of olive mill waste compost (C-OW) and tillage on GHG emissions and the NECB. Since these effects could be time-dependent, we considered measurements of GHG emissions made in the first and the third years after the application of the C-OW, corresponding to direct and residual impacts, respectively. Furthermore, since GHG emissions depend on the biological properties of soils, the activity of different enzymes was also studied to assess their effects on GHG emissions and *GWP*.

## 2. Materials and Methods

### 2.1. Site and Experiment Description

A field experiment was designed in the irrigated area of the Guadiana River, in the southwestern area of Spain (Lat: 38°55′58.14″ N, Long: 6°57′13.42″ O; Datum ETRS89). The location described above is an area traditionally used for growing rice under flooding systems, with an age of 11 years. According to the classification reported in [32], this area has a subtropical Mediterranean climate, and the mean values of rainfall and air temperature registered at field locations during the rice growing period in 2015–2017 are presented in Figure 1. This study was carried out between the 2015 and 2017 rice harvests, and crop and soil monitoring were performed between May and October each year. The rice fields began to be irrigated in the first days of May and the sprinkler irrigation ended in mid-September when the rice crop finished the days of the cycle with the full maturity of the plant.

A total of 3256 m^2^ of soil surface was used; each plot had an area of 10 × 18 m (180 m^2^), and each plot was separated from the remaining plots by a corridor of almost 2 m. Alperujo compost (C-OW) obtained from the Los Pedroches olive cooperative in Cordoba, Spain was used. The main C-OW characteristics were as follows: total organic carbon, 382 g kg^−1^; total nitrogen, 21.7 g kg^−1^; pH, 7.71; electrical conductivity, 2.32 ds m^−1^; and C:N ratio, 17.6.

### 2.2. Treatments

The treatments considered here were:

(1) With tillage (T);

(2) No tillage—direct seeding techniques (DS);

(3) Application of C-OW followed by tillage (TC);

(4) Application of C-OW followed by direct seeding (DSC).

An amount of 80 Mg ha^−1^ of C-OW was applied at the beginning of the study, in the month of April of the year 2015; it was incorporated to a depth of 15–20 cm using a harrow of discs.

Irrigation was applied to the rice through sprinklers, adding a total of each year of 10803 m^3^ ha^−1^, 8625 m^3^ ha^−1^, and 10309 m^3^ ha^−1^ for the years 2015, 2016, and 2017, respectively. The variety of rice used was Gladio, which belongs to the subspecies Indica; the rice was sown at a dose of 160 kg ha^−1^ with a Semeato TDNG 320 sowing machine.

### 2.3. Crop Management

The preparation of the land prior to sowing was carried out only in the T and TC treatments by tillage with a cultivator to a depth of about 15–20 cm, followed by the application of the seed to the ground. The sowing period coincided within the range of the first 5 days of May of each year [33]. Days before sowing each year, a complex bottom fertilizer was applied at a dose of 49.5 kg N ha^−1^, 99 kg P_2_O_5_ ha^−1^, and 148.5 kg K_2_O ha^−1^ (Fertiberia complex fertilizer) in the most important periods of crop development. Two top dressing fertilizations were performed, with urea applied at a rate of 80.5 kg of N ha^−1^ at each application.

### 2.4. Soil Analysis and GHG Sampling

The samples to determine the physical–chemical properties of the soil were taken at a depth of 0–20 cm (Appendix A), of which the total organic carbon (TOC) variable was measured by the wet oxidation method, with potassium dichromate and a subsequent evaluation of excess ferrous ammonium sulfate [34]. The water-soluble organic carbon (WSOC) of the soil was determined by extraction with distilled water in a proportion of 1/100 (p/v), and then a partial oxidation of the carbon was carried out with 1N potassium dichromate in a sulfuric acid medium followed by quantification using spectrophotometry at = 590 nm [35]. Measurements of the pH, FA, and HA content of the soil were also carried out, as described by Sims et al. [35]. N (Kjeldahl) was also measured as described by Sánchez-Llerena et al. [4].

The enzymatic properties were obtained on soil samples collected at a depth of 0–10 cm (Appendix A). For the dehydrogenase (DH) activity, 1 g of soil was incubated for 20 h at 20 °C in the dark with 0.20 mL of 0.4% 2-(4-iodophenyl)-3-(4-nitrophenyl)-5-phenyl-2H-tetrazolium chloride (INT) as the substrate. The β-Glucosidase (GL) activity was determined by incubating 1 g of soil with 4 mL of 25 mM 4-nitrophenyl-β-d-glucopyranoside in 0.1 M modified universal buffer (MUB) with a pH of 6.0. To assay the urease (UR) activity, 2 mL of 0.1 M phosphate buffer with a pH of 7.0 and 0.50 mL of 1.07 M urea were added to 0.50 g of soil and incubated for 1.5 h at 30 °C [36].

Static polyethylene chambers were used to assess the greenhouse gas emissions. At the time of sampling, the chambers were closed and a 20 mL air sample was taken from the headspace air inside the chamber at 0 and 30 min after closure. The air samples were stored in 30 mL airtight glass vials and were analyzed to quantify the concentration of nitrous oxide, methane, and carbon dioxide using the Agilent 5973 mass detector. All the quantification processes are described by Fangueiro et al. [24].

### 2.5. Net Carbon Balance in Aerobic Environments of Rice (NECB)

To estimate the SOC stock change under different management systems of soil, the NECB was calculated using the difference between the C input and output [37,38].
NECB = ∑ input (NPP + fertilizer + rice straw) − ∑ output (harvest C removal + respired C loss)

The C input sources included the net primary production (NPP) of rice, where the NPP means the total C fixation by plant biomass through photosynthesis, fertilizer, and the applied C-OW. The harvest removal and the respired C losses were considered as C output sources. In two different treatments, the amendment C-OW input was calculated using the compost application amount and its associated C content. The fertilizer C input was calculated using the C content of nitrogen (0.200) contributed by urea complex fertilizer. The respired C loss refers to heterotrophic respiration, which comes from SOC mineralization via CO_2_ and CH_4_ emissions from soils. The gas fluxes were investigated using the static chamber method, which was described above.

The NPP of rice was calculated by integrating the C uptake of each biomass part [39].
NPP (kg C ha^−1^) = NPPgrain + NPPstraw + NPProot + NPPlitter + NPPrhizodeposit

The NPP of the aboveground biomass was calculated by multiplying each biomass yield and its C content at harvesting time. The NPP of the root biomass was estimated as 10% of the NPP of the aboveground biomass [40]. The NPP of litter was assumed to account for 5% of the total biomass (aboveground and belowground biomass) NPP [41]. The NPP of the rhizodeposit was assumed to be 15% of the total biomass NPP [42]. The C input from the C-OW addition was calculated by multiplying the applied C-OW weight and the total C content.

### 2.6. Estimation of Global Warming Potential, Crop Yield, and GHG Intensity

Just before starting to collect the gas samples, the outside and inside temperature of the static chambers was recorded.

Fluxes of N_2_O, CO_2_, and CH_4_ (*F*, m^3^ m^−3^ min^−1^) were calculated as follows:
(1)$$F=\left(C_{t30}-C_{t0}\right)/30$$
where *C* (m^3^ m^−3^) is the gas concentration at time *t* (0–30 min).

The emission rates of the GHG´s (CH_4_, CO_2_, and N_2_O) (RE, g C or N ha^−1^ day^−1^) for each sample were calculated using the following Equation (2). The cumulative emissions were expressed in kg of carbon and nitrogen (i.e., CO_2_-C, CH_4_-C, and N_2_O-N):
ER = *F* × *M*/*V* × (273 + *T*/273) × *h* × *k* 10000(2)
where *F* is the gas emission flux calculated above (ppm min^−1^), *M* is the gas molecular weight (44 g mol^−1^ for CO_2_ or N_2_O and 16 g mol^−1^ for CH_4_), *V* is the volume of an ideal gas (0.022 m^3^ mol^−1^), *T* is the temperature during the sampling period (in °C), *h* is the height of the chamber (m), and *k* is the time corrected for a 1-day duration (1440 min).

To check the linearity of the increase in the gas concentration inside the chambers, air samples inside the chambers were taken at 0, 10, 20, 30, 40, 50, and 60 min [37] at the beginning of the experiment. This methodology was carried out in each treatment. Observing the results obtained, a final time of 30 min closure was chosen for all samples.

The global warming potential (*GWP*) was calculated based on Equation (3) [43], as follows [44]:
*GWP* = *con* CO_2_ + *con* CH_4_ × 28 + *con* N_2_O × 265(3)
where *con* is the concentration of each gas (CO_2_, CH_4_, and N_2_O) analyzed throughout the experiment (Mg CO_2_ eq ha^−1^).

The yield-scaled GHG emission (*GWPr*) (kg CO_2_ eq kg^−1^)) was calculated using the *GWP* and the *rice yields* reported by Peña et al. [33].(4)$$GWPr=\frac{GWP}{\mathrm{rice}\mathrm{yield}}$$

### 2.7. Statistical Analysis

The results were analyzed by analysis of variance (a two-way ANOVA) with repeated measures on the “treatment” factor and the “year” factor for the GHG, *GWP*, and *GWPr* emission data. IBM SPSS statistics software, package version 25.0, was used. All pairwise multiple comparisons were performed using the Duncan test. The Pearson correlation coefficient was used to study possible correlations between different parameters. Differences were considered statistically significant at a *p*-value of less than 0.05.

## 3. Results and Discussion

### 3.1. Emissions Rates of CO_2_-C and CH_4_-C

The emissions of CO_2_ and CH_4_ from the soil are important fluxes of C in the ecosystem and can represent 60 to 90% of the total ecosystem respiration [45]. The Table 1 shows the cumulative emissions of CO_2_-C, CH_4_-C, and N_2_O-N (kg ha^−1^) obtained with the different treatments considered here throughout the rice cycle and during the three years of the study.

In treatments DS and T (with no compost amendment), CO_2_-C emission rates of 3703 and 3855 kg ha^−1^, respectively, were obtained during the first year. These rates are higher than those reported by Lee et al. [46] from 2355 to 3246 kg C ha^−1^ in the first year of a study where they considered similar treatments. Nevertheless, Fangueiro et al. [24], in a study with tillage management and sprinkler irrigation performed in the same region as the present one, observed accumulated CO_2_-C emissions of 3103 kg C ha^−1^, in the same range of values as those obtained here.

In the first year of the study, the application of C-OW caused an increase in the net CO_2_ emissions of DSC and TC, by 1.57 times higher in DSC compared to DS and 2.13 times higher in TC versus T. In tilled management (T and TC), there was a significant increase in CO_2_ emissions (Table 1), probably due to the release of most of the C applied with the compost as well as a potential priming effect induced by the compost, which stimulated the release of soil-derived carbon [47]; however, CO_2_-C emissions accounted for 17.8% in tilled management for the first year and 15.6% in direct seeding management of the total C input on the ground (Table 2). The increase in the mineralization of the original organic matter of the soil (positive priming effect) was evidenced by an increasing microbial activity and respiration, resulting in an increase in CO_2_ emissions. The less intensive tillage used in the conservation agriculture management (DSC) was a key factor in the reduction of CO_2_ emissions, since DSC led to 2388 kg CO_2_-C ha^−1^ fewer net emission compared to the TC treatment in the first year following compost application. Lu et al. [48] also concluded that the practice of conservation agriculture inhibited CO_2_ emissions, which was due to the solidification of carbon with the increase in soil aggregation.

No significant differences were observed in terms of the CO_2_ emissions between the first and last years of the experiment for the two unamended treatments (DS and T) (Table 1). Regarding the residual effect of the C-OW in the last year of study, the management provided by this amendment during the first year reduced the CO_2_-C emissions by 54 and 58% for DSC and TC, respectively; however, DSC led to lower emissions of 1417 kg ha^−1^ of CO_2_-C compared to TC management. This decrease is in agreement with those reported by other authors, such as Liu et al. [49], who explained that zero tillage, compared to traditional tillage, improved soil aggregation and the soil organic carbon concentrations associated with aggregates with less disturbance and a higher crop residue input, which promoted carbon solidification due to lower microbial activity in the soil. In the present study, the values obtained for the microbial activity in the soil confirmed such a hypothesis, specifically for the DH (Appendix A), in which a significant reduction in the activity of 73% was observed for 2017 compared to 2015 in the DSC treatment. However, the accumulated emissions by the TC treatment in 2017 presented values of up to 5182 kg ha^−1^ CO_2_-C, and this emission rate could be attributed to the availability of the substrate and the improvement in microbial activity, since it was obtained in the residual year. An increase in the GL of 9.56 times with respect to the GL of T (Appendix A) could be the reason for a consequent increase in CO_2_ emissions under conventional tillage [50]. Table 1 shows the values of the cumulated CH_4_-C emissions obtained in each treatment in each of the three years considered here. Several chemical, physical, and biological factors regulate the production and oxidation of CH_4_ in rice soils [51]. Regardless of the management system used and the application or not of C-OW, it was observed that there were negative cumulative emission rates for the years 2016 and 2017. Average values of −5.95 kg of CH_4_-C ha^−1^ were obtained for all the years and management types of the study (Table 1). Therefore, under aerobic irrigation conditions, it can be concluded that the soil serves as a sink for CH_4_, because the balance between CH_4_ production by methanogen microorganisms and the consumption by methanotrophs is negative [52]. Similar results were found by Kreye et al. [53], who also obtained negative emission rates, showing the sink effect in aerobic rice systems. However, our management with aerobic irrigation caused a lower cumulative emission than traditional anaerobic rice management in studies carried out in the same area and during the same dates. Fernández-Rodríguez et al. [54] obtained cumulated CH_4_-C emission values that were 54 times higher than those shown in our study with aerobic irrigation conditions. The drying period inhibits CH_4_ production and leads to an increase in the O_2_ concentration in the soil due to the shorter duration of the anaerobic environment [55]. Under aerobic irrigation, CH_4_ emissions are reduced by up to 99% compared to anaerobic irrigation systems; therefore, the type of irrigation is a crucial factor in the emissions of this gas [24]. Similarly, other authors have stated that the intermittent irrigation system (wetting–drying) significantly reduces CH_4_ emissions in rice cultivation [56]. García-Marcos et al. [57] also demonstrated the sink capacity in terms of CH_4_ emissions, reporting average values of −0.048 kg of CH_4_ between tillage and non-tillage handling in the Triticale (x *Triticosecale*).

### 3.2. Emissions Rates of N_2_O-N

There are two processes that promote the production of N_2_O emissions: nitrification and denitrification. Table 1 shows the values of cumulated N_2_O emissions for the different treatments studied in the three different years. In the first year of the study (2015), no significant differences were observed between the crop management treatments (DS and T), with an average value of 17.3 kg of N_2_O-N ha^−1^. This value is much higher than the one of 3.91 kg of N_2_O-N ha^−1^ reported by Fernández-Rodríguez et al. [54] in a study with traditional flooded rice. It can then be concluded that the cultivation of rice under aerobic irrigation systems increases the emission of N_2_O compared to anaerobic systems [58,59]. The productions of N_2_O increase due to the saturation of moisture soil environs. Soil moisture variations with subsequent moist and dry periods enhance the production of N_2_O [60]. This occurs due to an increase in microbial functioning in humid environments. In our study, T values that were 1.86 times higher than those shown by other authors were obtained for rice flooding conditions (Appendix A) [54]; therefore, microbial activity is inhibited under very high humidity levels. Soil humidity and temperature, as well as the application of nitrogenous fertilizers, are the main drivers of N_2_O emissions in soil [47,61]. Other authors [62] obtained values near 17.8 kg of N_2_O-N ha^−1^, similar to those obtained here (16.1 kg of N_2_O-N ha^−1^). In aerobic soils with nitrogen fertilizer applications, the N_2_O rate may be favored with an increase in nitrogen fertilization application [63]. However, the use of high contributions of nitrogenous fertilizers in areas of intensive agriculture to produce higher yields is very normalized [64]. Regarding the values shown in Table 1 for the year 2016, a significant decrease in N_2_O emissions was observed for the DS and DSC managements of 64% and 39%, respectively, compared to the year 2015, while in the tilled management groups, there was no significant difference with respect to the values of the initial year. In the treatments with conservation agriculture, decreases in the peaks caused by the background nitrogen fertilization at the beginning of the crop cycle were observed. As a result, a lower concentration of N_2_O-N accumulated throughout the rice cycle, which may have been a negative effect of the high rainfall that occurred in the first months of the crop cycle (May and June) and which caused a drop in the temperatures compared to the same months in 2015 and 2017, slowing down and delaying the nitrification and denitrification processes. Other authors [65,66] have shown that the emissions of N_2_O depend largely on the availability of oxygen and on the temperature in the soil. As Table 1 shows, in the third year of the study, a significant and generalized increase in N_2_O emissions was observed for all management groups, and even more so, if possible, a greater increase was observed in the management groups with C-OW, DSC, and TC, with respect to their counterparts without amendment. The increase in N_2_O emissions in 2017 offered higher values for non-till management (DS and DSC) and tilled management (T and TC) of up to 1.94 and 1.49 times, respectively, compared to 2016. Higher temperatures were recorded in 2017 relative to 2016 in the period where the bottom and cover fertilizers were applied (May, June, and July). Hasanah et al. [67] observed in experimental fields that the highest N_2_O emission rates occurred in the 32.6 °C to 33.8 °C range of temperatures, and values very close to the average of 31.5 °C were obtained in our field experiment in 2017.

In the last year of the experiment, no significant differences were observed between all the treatments, although a general increase in N_2_O emissions was observed in the tilled treatments (T and TC), as much as 1.36 times higher than the accumulated N_2_O-N in T compared to DS, and a 1.32-times higher N_2_O-N was observed TC versus DSC. Conventional tillage disrupts the soil structure, increases the temperature, and oxidizes the soil organic matter, which can induce N_2_O emissions from the soil as compared to no tillage [50]. In line with our results, several studies have reported the importance of the mineral nitrogen dosage combined with different tillage methods on N_2_O emissions [68,69]. As a result, the dominance of soil mineralization and amendments with a low C:N ratio (< 30) can influence the availability of “immobilized” the N that is ready to be absorbed by plants or microbes. The existence of organic amendments (such as C-OW) in a high C:N soil surface can enhance the “lock-in” of useful N for bacterial availability, thus reducing denitrification reactions and the carbon dioxide and N_2_O emissions [70]. The WSOC of the soil is used as a labile C resource for microbial growth, which causes a higher rate of N_2_O-N emissions and which affects the representation of this gas [49]. Li et al. [71] have already shown slightly higher values of soil temperature with tillage management versus non-tillage management, and have also obtained positive correlations in terms of the soil temperature and N_2_O emissions.

### 3.3. Net Carbon Balance in Aerobic Environments of Rice (NECB)

The carbon content of a soil is important in Mediterranean systems based on intensive cultivation, which, if not addressed, leads to notable losses of the soil organic matter content, especially in arid and semi-arid areas where soils are often already very poor in organic matter content [72]. Table 2 shows the values of the NECB obtained at the end of the experiment. The NPP is defined by the quantity of photosynthates that are not used for respiration, and are therefore available for other processes; the values include a 75–80% contribution of the aboveground biomass of the rice crop, while the underground biomass represents 20–25%. In the first year of the experiment, the NPP value did not offer significant differences between the four treatments (DS, DSC, T, and TC), showing an average value of 19,238 kg of C ha^−1^. Similarly, Xia et al. [73] obtained 19,619 kg of C ha^−1^ with an application of 240–300 kg of N ha^−1^. Good agricultural management is essential to increase biomass productivity, thus increasing the SOC content in farmland as well as improving crop yields to strengthen food security [74]. The NECB was positive for all treatments in the first year, which indicates a higher carbon content in the soil at the end of the rice campaign. Treatments receiving C-OW obtained an average of 27,534 kg of C ha^−1^ in the first year; however, the DS and T management groups obtained an average of 416 kg of C ha^−1^. This increase in DSC and TC management was due to the incorporation of C-OW, which represented an amount of 30,560 kg of C ha^−1^. The application of the C-OW had the objective of mitigate the three years of lack of organic matter in direct sowing management and increasing water retention in the soil. Studies carried out by López-Piñeiro et al. [75] used a C-OW amendment with a total organic carbon (TOC) content of 383 g kg^−1^, similar to the TOC of our C-OW amendment (382 g kg^−1^), added to intensively cultivated soils. In addition, in studies similar to ours, authors such as Jeong et al. [31] used composted cattle manure with TOC values of 359 g kg^−1^ in an experiment where they applied 2 Mg ha^−1^, and obtained results for the NPP of 8489 kg of C ha^−1^. In the first year (direct effect), the entry of C into the treatments by the C-OW resulted in an increase of 550 kg of C in each repetition of the amended treatments (180 m^2^).

The managements modified with C-OW, DSC, and TC (Table 2) in the year of the direct effect obtained higher values than the DS and T administrations for the C output, with values of 11% and 15%, respectively. Lee et al. [46] also obtained a significant increase of 30% in the C outputs due to the application of straw in the first year, with respect to the treatments where no straw was applied. This was mainly due to the contribution of carbon from the C-OW, but this effect caused an increase in the CO_2_-C emissions of the DSC and TC treatments compared to their DS and T counterparts, with an increase in emissions of 57% and 113%, respectively, and with a significant difference in the management of TC compared to the management of T, which could be due to a greater availability of carbon from the microbial biomass, carbohydrates, and organic acids [76]. The application of tillage with C-OW in the soil for TC management could explain the significant increase found in the C outputs of 10% (Table 2), with respect to DSC management. This could be due to the fact that C outputs from the soil are generated by heterotrophic respiration by microorganisms, autotrophic emissions by the roots, and methanogenic bacteria, in which the microorganisms use substrate sources such as carbon, which is more available due to the incorporation caused by the tillage [77].

In the third year (Table 2), the residual effect showed us an average C input value for all treatments of 16,269 Kg of C ha^−1^, but no significant differences were observed between the different treatments, although there was a slight rise for the TC management of 1.18 times above the mean of the other managements (DS, DSC, and T). The mean values of the residual year of entry of C were below those obtained in the first year (direct effect); this was due to the fact that the application of C-OW was only incorporated in the first year; therefore, it increased the mean C input values obtained in all treatments, and more specifically, in the treatments with amendment (DSC and TC).

The outputs of C, in the form of CO_2_-C, showed average values of 3396 kg of C ha^−1^, in the original DS and T managements. However, the residual effect values in the TC treatment were 1.42 times higher than in its T counterpart, and were also 1.38 times higher than in the DSC treatment. These results could explain how in the third year, the effect of no tillage in rice cultivation caused a lower C output in the form of CO_2_-C [77]. In addition, the application of C-OW helped to increase the values of the C output compared to not tilling with C-OW.

During the last year, the NECB showed positive values in the DS treatment (Table 2), indicating that a positive result was produced in the accumulations of C in the management group, resulting in the storage of up to 189 kg of C ha^−1^. The T management, although it did not obtain a significant difference, showed negative values (−130 kg of C ha^−1^), which turned out to be more of an emitter of C to the environment than an accumulator of C in the soil, as demonstrated by Thapa et al. [78] obtaining 53% more accumulation of C ha^−1^ in non-tillage soils compared to soils tilled in strips. The amended DSC and TC treatments had NECB values of −661 and −1268 kg of C ha^−1^, respectively, which could increase the enzyme activity of the soils [33]. This respiration was promoted by the application of C-OW OW in the first year of the experiment, demonstrating that a high soil and air temperature shows a quadratic response with the outputs of C ha^−1^ [78].

### 3.4. Global Warming Potential (GWP) and GWP Yield (GWPr)

The effect of the application of organic matter (C-OW) combined with the use of direct sowing and traditional tillage on the *GWP* was determined. For the year 2015, Table 1 indicates a range of values between 19.9 and 38.1 Mg CO_2_ eq ha^−1^. These results are similar to those obtained in other experimental fields, where spring cereals were grown with differences in mineral fertilizer techniques and organic fertilizers and the researchers obtained variations from 25.6 to 33.8 Mg CO_2_ ha^−1^ [79]. It should be noted that in our study, the chemical fertilizer was applied to all the treatments. The soils where C-OW (DS and T) were not applied obtained average values of 21 Mg CO_2_ eq ha^−1^, without showing a significant difference between them; however, a very pronounced increase was observed in the management of TC with respect to its counterpart T, with the *GWP* being up to 72% times higher. Other studies have indicated a scaled increase in the *GWP* of 56 and 163% for cultivated rice fields with the application of manure and straw compared to a mineral amendment [80]. This could be because the application of manure residues and straw provides abundant sources of fresh C, which promotes *GWP* production in aerobic paddies [81]. In addition, the TC treatment obtained higher scaled *GWP* values than the DSC management (1.33 times higher). It has been shown in cereal crops with the incorporation of organic amendment with residues of straw from the previous year that the application of no tillage causes a lower *GWP* of up to 1.32 times, compared to management with tillage, which demonstrates that the alteration of soil with working machinery has a significant effect on global warming [82]. As indicated in Figure 2, the contribution of CO_2_ gas to the *GWP* of the total treatments was on average 71% between the managements with an application of CO-W and those without CO-W. A 16% greater contribution was observed for the treatments with CO-W, this increase being normal due to the effect caused by CO-W of a higher COT content in the soil. The contribution of N_2_O gas to the *GWP* showed lower percentages than those contributed by CO_2_ gas in all handlings. However, the management without CO-W (DS and T) obtained a greater representation of N_2_O over the *GWP* of 34%, while the treatments with C-OW obtained a representation of 23% of N_2_O over *GWP* (Figure 2). This decrease in the contribution of N_2_O may have been favored by the low C:N ratio obtained by the managements in which a CO-W amendment was applied. This has been demonstrated by the results obtained in other studies, where the N_2_O emissions were negatively correlated with the C:N ratio [83].

The residual effect of the treatments showed values between 20 and 31.1 Mg CO_2_ eq ha^−1^; these results do not demonstrate significant differences with respect to the effect of the *GWP* in the first year. However, the CO-W treatment in no-till management (DSC) caused a decrease in the gases affecting the *GWP* compared to the first year (25%), which brought the *GWP* (DSC) values closer to the original DS management. This could be the reason for the decrease in the TOC content in the DSC treatment and the no-till regime for 3 years, which caused a more compact physical condition of the soil, thus avoiding a light flow of gas emissions. In the TC treatment, a non-significant decrease in the *GWP* values was also observed with respect to the initial year. In addition, the TC management maintained a greater *GWP* content with respect to the T management in the year of the residual effect; however, this increase was reduced compared to the first year, since it went from having a significant difference of 72% in the first year to a significant difference of 25% in the second year, which may have been due to a significant decrease in the content of WSOC in the third year with respect to the first year, by 79% (Table 1). The content of WSOC is a resource for microbial activity that favors the gases that participate in increasing the *GWP* [84]. In the residual year, the contribution of each gas to the average *GWP* of the total number of treatments was 58% for CO_2_ and 42% for N_2_O, while CH_4_ continued its activity of capturing this gas. Regarding the original managements, DS and T showed very similar values (57% and 53%, respectively), while the managements with an application of CO-W, DSC and TC, showed values 4% or above the mean of the total of the treatments. Regarding the contribution of N_2_O gas to the total *GWP*, the values did increase in the residual year with respect to the initial year, by 1.48 times in general for all treatments, without there being a large gap between the different treatments (DS, DSC, T, and TC). However, the original treatments, DS and T, obtained, on average, a higher contribution of N_2_O to the *GWP* than their counterparts, DSC and TC, by 12%. A positive correlation of the WSOC with *GWP* emissions was observed (r = 0.477, *p* < 0.05). Therefore, this effect could explain the increase in the WSOC content originating in the residual year compared to the initial year for the DS and T managements (50% and 145%, respectively).

Regarding the effect of the application of C-OW, it was observed that during the first year of study (2015, direct effect), there were significant increases. Thus, the *GWP*r in DSC increased with respect to DS by 1.41 times, and in TC, it increased with respect to T by 1.89 times (Table 1). In a meta-analysis carried out by Zhao et al. [85] after reviewing 230 publications, it was indicated how the application of organic amendments in rice cultivation could cause an average increase of 37.3% in the *GWP*r values. Three years after the application of the compost (residual effect), the value of the *GWP*r in the treatments that received the amendment was higher than that registered in the original treatments. Thus, the *GWP*r in DSC increased with respect to DS by 1.24 times, and in TC, it increased with respect to T by 1.14 times (Table 1). However, unlike the direct effect, in the residual effect the increases were not significant. These results show how, in the medium term, the application of organic materials under aerobic irrigation systems results in lower *GWP*r values compared to anaerobic irrigation systems [33], probably due to differences in the mineralization of organic matter from the amendment. In fact, the correlation study shows that there is a significant (*p* < 0.05) and negative correlation between the *GWP*r and HI (r = −0.465). Therefore, authors such as Thangarajan et al. [47], with the aim of reducing *GWP*, recommend using stabilization processes, such as composting, to transform easily degradable compounds into stable organic matter.

## 4. Conclusions

The use of sprinkler irrigation in combination with a C-OW application under different tillage systems leads to important changes in the soil properties (organic matter, pH, and N), which may affect the GHG emissions. Thus, under sprinkler irrigation, the implementation of direct seeding rather than tillage managements could be an interesting alternative to reduce the emissions of CO_2_ and N_2_O from rice soils. Furthermore, regardless of treatments, the use of sprinklers is an optimal irrigation management strategy in order to reduce the emissions of CH_4_, which is considered one of the major contributors to the global warming potential. The effects of a C-OW application on the *GWP* were time-dependent, probably due to the aging process. Thus, whereas the application of C-OW led to significant increases in the first year (direct effect), regardless of the tillage system, in the third year (residual effect), there were no significant differences between direct seeding treatments. Therefore, the combination of sprinkler irrigation with direct seeding and a C-OW application could be a viable technique for rice crops in order to reduce the GHG emissions and *GWP* under Mediterranean conditions.

## Figures and Tables

**Figure 1 plants-11-03454-f001:**
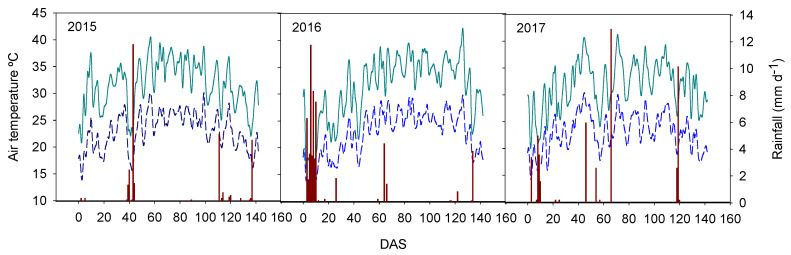
Mean rainfall and air temperature pattern registered at field locations during the rice growing period in 2015–2017. DAS: days after seeding. (-) maximum temperature air, (-) medium temperature air and (-) rainfall.

**Figure 2 plants-11-03454-f002:**
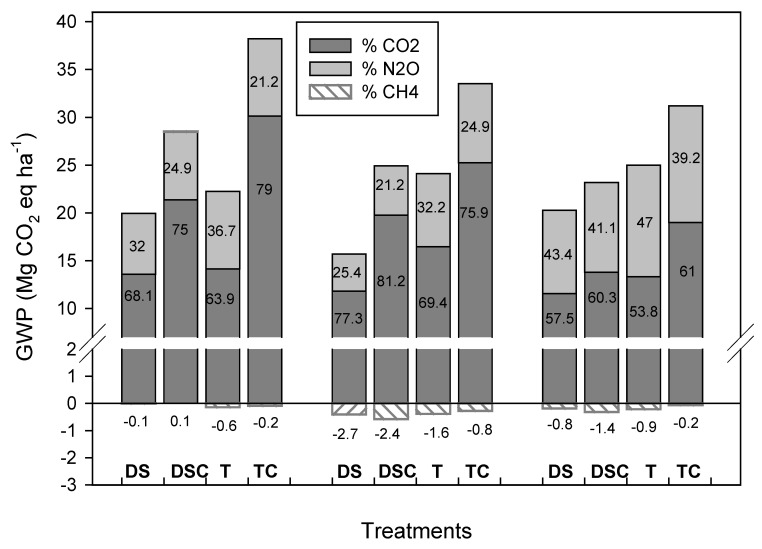
Contribution of each gas to the *GWP* for the different managements (%). Rice cultivated by: DS, direct seeding; DSC, direct seeding compost; T, tillage; and TC, tillage compost. *GWP*: global warming potential.

**Table 1 plants-11-03454-t001:** Effect of the different management systems on the cumulative emissions of CO_2_-C, CH_4_-C, and N_2_O-N (kg ha^−1^) during the rice cultivation cycle.

	CO_2_-C (kg ha^−1^)	CH_4_-C (kg ha^−1^)	N_2_O-N (kg ha^−1^)	*GWP*	*GWPr*
2015					
DS	3703 aA	−0.270 aB	15.3 aB	19.9 aB	2.27 aA
DSC	5832 aB	1.05 aB	17.1 aB	28.5 bA	3.22 bA
T	3855 aA	−4.41 aA	19.5 aA	22.1 abA	2.27 aA
TC	8220 bA	−2.50 aA	19.4 aA	38.1 cA	4.30 cA
2016					
DS	3222 aA	−10.8 aA	9.29 aA	15.3 aA	2.31 aA
DSC	5391 abB	−15.5 aA	12.3 abA	24.3 abA	5.34 bB
T	4491 aA	−10.3 aA	18.3 abA	23.7 abA	4.50 abC
TC	6892 bA	−7.29 aA	19.9 bA	33.3 bA	5.80 bA
2017					
DS	3153 aA	−3.45 abB	20.6 aC	20.0 aB	2.74 aA
DSC	3765 aA	−8.46 aAB	22.2 aC	22.7 aA	3.42 abA
T	3639 aA	−5.79 abA	28.0 aB	24.8 aA	3.18 abB
TC	5182 bA	−1.61 bA	29.3 aA	31.1 bA	3.63 bA
Y	8.30 **	6.44 **	25.7 ***	3.66 *	17.9 ***
T	14.3 ***	NS	NS	13.2 **	8.45 **
Y x T	NS	NS	NS	NS	3.35 *

Rice cultivated by: DS, direct seeding; DSC, direct seeding compost; T, tillage; and TC, tillage compost. *GWP*: global warming potential; *GWPr*: global warming potential yield. ANOVA factors are: Y, year; T, treatment; and Y x T, interaction of year * treatment. F-values indicate the significance levels: * *p* < 0.05; ** *p* < 0.01; *** *p* < 0.001; and NS, not significant. Different letters indicate differences (*p* < 0.05) between treatments in the same year (lower-case letters) and between years within the same treatment (upper-case letters).

**Table 2 plants-11-03454-t002:** Effect of rice grain productivity and net carbon balance in aerobic environments of rice on soils amended with composted olive mill wastes during investigation period.

Year	1st Year				3rd Year			
Treatments	DS	DSC	T	TC	DS	DSC	T	TC
C input (kg C ha^−1^)	19020 a	49578 b	20435 a	49167 b	15683 a	14532 a	16463 a	18399 a
NPP	18988 a	18986 a	20403 a	18575 a	15651 a	14500 a	16431 a	18367 b
Grain	8785 a	8855 a	9699 a	8776 a	7308 ab	6782 a	7698 ab	8581 b
Straw	6191 a	6120 a	6394 a	5874 a	5037 a	4655 a	5262 a	5906 a
Root	749 a	749 a	805 a	733 a	617 a	572 a	648 a	724 a
Litter	786 a	786 a	845 a	769 a	648 ab	600 a	680 ab	761 b
Rhizodeposit	2477 a	2476 a	2661 a	2423 a	2041 ab	1891 a	2143 ab	2396 b
Fertilizer (urea)	32	32	32	32	32	32	32	32
C-OW	0	30,560	0	30560	0	0	0	0
C output (kg C ha^−1^)	18679 a	20808 a	19944 a	22868 a	15494 a	15194 a	16593 a	19667 a
Harvest removal	14976	14975	16093	14650	12345	11437	12960	14487
CO_2_-C	3703 a	5832 a	3856 a	8221 b	3153 a	3765 a	3639 a	5182 b
CH_4_-C	−0.270 a	1.05 a	−4.41 a	−2.50 a	−3.45 ab	−8.46 a	−5.79 ab	−1.61 b
NECB	341	28,770	491	26298	189	−661	−130	−1268

Rice cultivated by: DS, direct seeding; DSC, direct seeding compost; T, tillage; and TC, tillage compost. ANOVA factors is: T, treatment. Different letters indicate differences (*p* < 0.05) between treatments in the same year (lower-case letters).

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
