# Peer review of "Effects of Combined Use of Olive Mill Waste Compost and Sprinkler Irrigation on GHG Emissions and Net Ecosystem Carbon Budget under Different Tillage Systems"

_plants, 2022, doi:10.3390/plants11243454_

Round 1

Reviewer 1 Report

Check your manuscript for minor corrections. See the attached file please

Author Response

Dra. Milica Čudić

Editor

Plants

December 05 2022

Dear Dra. Milica Čudić

We want to thank the reviewers for your comments and helpful suggestions. We trust that the new version of the manuscript (plants-2075477) is in accordance with the reviewer´s reports.

Reviewer #1:

The comments of reviewer#1 have been addressed in the amended manuscript.

Reviewer 2 Report

Dear Authors,

the paper is definiately of interest for the potential readers of Plants and I believe many more journals.

GHGs emission is an emerging topic, as well as utilization of organic wastes (also byproducts) and there is a great need to develop strategies that will allow to humify rathen than mineralize SOC. Therefore, after reviewing Your paper (scientifically sound, transparent and with conclusions supported by the results) I recommend its acceptance after minor revision.

Corrections I suggest are rather cosmetic, but please pay attention to some spelling mistakes (i.e. line 34 decreased -->decrease),  logic construction of some sentences (Line 60 -->the sentence is unclear/unfinisherd?). Additionally, please change the refering manner i.e. Instead of writing: [XX} refers that... use Surname et al [XX] as it is much more transparent and easier to follow that way.

Author Response

Dra. Milica Čudić

Editor

Plants

December 05 2022

Dear Dra. Milica Čudić

We want to thank the reviewers for your comments and helpful suggestions. We trust that the new version of the manuscript (plants-2075477) is in accordance with the reviewer´s reports.

Reviewer #2:

The paper is definiately of interest for the potential readers of Plants and I believe many more journals.

GHGs emission is an emerging topic, as well as utilization of organic wastes (also byproducts) and there is a great need to develop strategies that will allow to humify rathen than mineralize SOC. Therefore, after reviewing Your paper (scientifically sound, transparent and with conclusions supported by the results) I recommend its acceptance after minor revision.

- Corrections I suggest are rather cosmetic, but please pay attention to some spelling mistakes (i.e. line 34 decreased -->decrease), logic construction of some sentences

As suggested by the reviewer “decreased” replaced with “decrease” in the amended manuscript

(Line 60 -->the sentence is unclear/unfinisherd?).

As suggested by the Reviewer, this sentence has been rewritten in the amended manuscript.

Additionally, please change the refering manner i.e. Instead of writing: [XX} refers that... use Surname et al [XX] as it is much more transparent and easier to follow that way.

As suggested by the Reviewer, the referring manner has been changed in the amended manuscript.

Reviewer 3 Report

The manuscript relates to the “Combined use of olive mill waste compost and sprinkler irrigation on GHG emissions and net ecosystem carbon budget under different tillage systems”.

The text is generally well-written; the authors should check their manuscript for minor grammar issues. Some minor points are attached below. All answers should be included in the text.

1.      Abstract: should be shorter.

2.      Abstract: “…-C” should be explained when appearing for the first time

3.      L.287: Table 3?

4.      Table 2: Please, include data for the 2nd year as well.

5.      Given that the life cycle of Oryza Sativa L. is less than a year, please, describe the effect of new seeding. Could the successive seedings be the reason for the lack of nutrients and reduction of CO2-C with the years?

6.      Please, check the grammar in Ls. 371 and 372.

7.      Regarding the N2O-N increase during the third year of cultivations, are there other similar literature data that exceed the three-year study? Is the N2O-N still high?

Author Response

Dra. Milica Čudić

Editor

Plants

December 05 2022

Dear Dra. Milica Čudić

We want to thank the reviewers for your comments and helpful suggestions. We trust that the new version of the manuscript (plants-2075477) is in accordance with the reviewer´s reports.

Reviewer #3

The manuscript relates to the “Combined use of olive mill waste compost and sprinkler irrigation on GHG emissions and net ecosystem carbon budget under different tillage systems”.

The text is generally well-written; the authors should check their manuscript for minor grammar issues. Some minor points are attached below. All answers should be included in the text.

Abstract: “…-C” should be explained when appearing for the first time

As suggested by the Reviewer, the -C and -N forms has been explained in the amended manuscript.  

L.287: Table 3?

This mistake has been corrected in the amended manuscript. Thank you

Table 2: Please, include data for the 2nd year as well.

The C-OW was only applied in the 1st year of the study. Thus, we evaluated the direct (1st year) and residual (3rdyear) effects of C-OW on net ecosystem carbon budget. In fact, similarly studies such us xxxxx also determined the direct and residual effects of organic amendments on net ecosystem carbon budget in the year of the study.

Qi, L., Pokharel, P., Chang, S.X., Zhou, P., Niu, H., He, X., Wang, Z., Gao, M. Biochar application increased methane emission, soil carbon storage and net ecosystem carbon budget in a 2-year vegetable–rice rotation. Agric. Ecosyst. Environ. 2020, 292, 106831.

Mandal, U.K., Bhardwaj, A.K., Lama, T.D., Nayak, D.B., Samui, A., Burman, D., Mahanta, K.K., Sarangi, S.K., Mandal, S., Raut, S. Net ecosystem exchange of carbon, greenhouse gases, and energy budget in coastal lowland double cropped rice ecology. Soil Tillage Res. 2021, 212, 105076.

Belenguer-Manzanedo, M., Alcaraz, C., Camacho, A., Ibáñez, C., Català-Forner, M., Martínez-Eixarch, M. Effect of post-harvest practices on greenhouse gas emissions in rice paddies: flooding regime and straw management. Plant Soil, 2022, 474, 77-98.

Given that the life cycle of Oryza Sativa L. is less than a year, please, describe the effect of new seeding. Could the successive seedings be the reason for the lack of nutrients and reduction of CO2-C with the years?

The implantation of the new sowings in the spring season, are explained by the author Peña et al. [33], which at the reviewer's suggestion we have introduced the citation in section 2.3. Crop management.

Please, check the grammar in Ls. 371 and 372.

As suggested by the Reviewer, the paragraph of the introduction section has been rewritten in the amended manuscript.

Regarding the N2O-N increase during the third year of cultivations, are there other similar literature data that exceed the three-year study? Is the N2O-N still high?

Under the conditions where we conducted our aerobic soil studies with nitrogenous fertilizer applications, the N2O rate may be favored by increasing the application of nitrogenous fertilizers year after year. In 2017, higher temperatures were recorded in the period where bottom and cover fertilizers were applied (May, June and July). Hasanah et al., (2017) observed in experimental fields that the highest N2O emission rates occurred in the temperature range of 32.6ºC to 33.8ºC, values very close to the average of 31.5ºC obtained in our experiment of field in 2017. Also other authors such as Mascarenhas et al., (2020) showed an increase every year in the N2O rate in various growing seasons

Hasanah, N.A.I., et al. Int. J. Environ. Sci. Technol. 2017, 10, 206-214.

Mascarenhas, Y.S. et al. Pesqui. Agropecu. Bras. 2020, 55, 1-10.
